# Allosteric coupling of sub-millisecond clamshell motions in ionotropic glutamate receptor ligand-binding domains

Suhaila Rajab[1], Leah Bismin[1], Simone Schwarze[1], Alexandra Pinggera[2], Ingo H. Greger[2] & Hannes Neuweiler [1 ✉]

Ionotropic glutamate receptors (iGluRs) mediate signal transmission in the brain and are important drug targets. Structural studies show snapshots of iGluRs, which provide a mechanistic understanding of gating, yet the rapid motions driving the receptor machinery are largely elusive. Here we detect kinetics of conformational change of isolated clamshell-shaped ligand-binding domains (LBDs) from the three major iGluR sub-types, which initiate gating upon binding of agonists. We design fluorescence probes to measure domain motions through nanosecond fluorescence correlation spectroscopy. We observe a broad kinetic spectrum of LBD dynamics that underlie activation of iGluRs. Microsecond clamshell motions slow upon dimerization and freeze upon binding of full and partial agonists. We uncover allosteric coupling within NMDA LBD hetero-dimers, where binding of L-glutamate to the GluN2A LBD stalls clamshell motions of the glycine-binding GluN1 LBD. Our results reveal rapid LBD dynamics across iGluRs and suggest a mechanism of negative allosteric cooperativity in NMDA receptors.

[1] Department of Biotechnology & Biophysics, Julius-Maximilians-University Würzburg, Würzburg, Germany. [2] Neurobiology Division, Medical Research Council Laboratory of Molecular Biology, Cambridge, UK. ✉email: hannes.neuweiler@uni-wuerzburg.de

Signal transmission at excitatory synapses is mediated by ionotropic glutamate receptors (iGluRs) that are ubiquitously expressed in the central nervous system[1,2]. iGluRs are ligand-gated ion channels that play key roles in brain development and higher-order cognitive functions including learning and memory. Receptor malfunction contributes to various brain disorders such as epilepsy, stroke, Alzheimers disease, and schizophrenia[3]. Hence, iGluRs are prominent targets for drug development. Based on pharmacology and structural homology, iGluRs are divided into three major subtypes, namely AMPA, kainate, and NMDA receptors, all of which form trans-membrane spanning tetrameric assemblies[1]. Each receptor subunit is built of semi-autonomous domains that are connected by flexible linkers. A single subunit consists of the extracellular N-terminal- and ligand-binding domains, the trans-membrane ion channel, and the intracellular C-terminal domain (NTD, LBD, TMD, and CTD, respectively)[4]. The extracellular NTDs and LBDs are dimeric, bi-lobate structures resembling clamshells. The bi-lobate shape of the LBD is structurally similar to bacterial periplasmic amino acid-binding proteins[5]. The two extracellular iGluR domain layers, formed by the NTDs and LBDs, respectively, are arranged as dimers of dimers. Receptor activation is triggered by agonist binding to the LBD inter-lobe cleft. Clamshell closure is transferred to the gate through an upward rotation of the lower D2 lobe[6], triggering ion flux to depolarize the post-synaptic cell enabling signal transduction[7].

Over the past two decades, structural studies, involving x-ray crystallography and cryo-electron microscopy (cryo-EM), together with electrophysiology have provided detailed insights into the structure–function relationship of iGluRs[4,8]. Crystal- and cryo-EM structures of homo- and heteromeric iGluRs from AMPA[9–14], NMDA[15–19], and kainate[20] subtypes unveiled mechanisms of gating control. iGluR-mediated signal transduction is rooted in a complex network of conformational motions of individual domains, which are elusive to experimental observation. The dynamics of the individual domains, their inter-domain communication and modulation by binding of agonists and antagonists are beginning to be defined[21]. Single-molecule fluorescence resonance energy transfer (smFRET) studies have provided additional insights into the transitions between structural end states in solution and reveal their conformational heterogeneity[22].

A particular focus is on the LBD, the "muscle" of the receptor, with domain motions providing the trigger that initiates channel gating[23]. A wealth of crystal structures of isolated LBDs in complex with various ant/agonists show that the domain can adopt various conformations[3,24]. However, the extent to which these are populated in solution and their time scales of interconversion remain unclear. Molecular dynamics simulations suggests that the LBD populates a more extended ensemble of conformations than observed experimentally[25].

Gating kinetics differ between iGluR subtypes[8], and depend upon the nature of agonist within a subtype[26]. AMPA/kainate receptors deactivate within a few ms while NMDA receptors require hundreds of ms[2]. Rapid kinetics of receptor activation[1] suggest that the elementary LBD clamshell dynamics underlying gating are on a similarly fast ms or sub-ms time scale. Structural studies suggest that the amplitudes of LBD lobe motions are on the order of one nanometer (nm)[1,27,28]. This calls for high-resolution spectroscopic methods, which are sensitive on the sub-ms temporal and one-nm spatial scale, to detect functionally relevant lobe motions.

Here we designed fluorescent probes that, in combination with fluorescence correlation spectroscopy (FCS), fulfill these high-resolution requirements. We made use of an environmentally sensitive oxazine fluorophore that we labeled site-specifically to the mouths of isolated LBD clamshells from the three major iGluR subtypes, namely AMPA, kainate and NMDA. The label transforms conformational fluctuations of the LBD into fluorescence fluctuations that are detected by FCS[29]. We describe pronounced sub-millisecond fluctuations in the apo state of LBDs from all three subtypes and study their modulation on agonist binding and dimerization. We also reveal a pathway of allosteric communication in NMDA LBD dynamics across the dimerization interface.

## Results

**Sub-millisecond clamshell motions of iGluR LBDs.** In order to detect inter-lobe motions between the upper D1 and lower D2 lobes (clamshell conformational dynamics) we made use of the environmentally sensitive fluorophore AttoOxa11 (Atto-Tec) that reports changes in its micro-environment by changes in fluorescence emission intensity in the far-red spectral range. Fluorescence of AttoOxa11 is efficiently quenched upon formation of van der Waals contact with the side chain of tryptophan (Trp) through photoinduced electron transfer (PET) (Fig. 1a)[30]. Rapid conformational changes that are accompanied by formation and disruption of fluorophore/Trp interaction result in PET fluorescence fluctuations that can be detected by FCS (PET-FCS)[29,31]. We introduced AttoOxa11 in the D1 lobe of iGluR LBD subtypes by mutating a solvent-exposed side chain to cysteine (Cys) and modifying it with thiol-reactive AttoOxa11. The natural quencher Trp (W) was introduced in the D2 lobe (Fig. 1b–d). The labeling

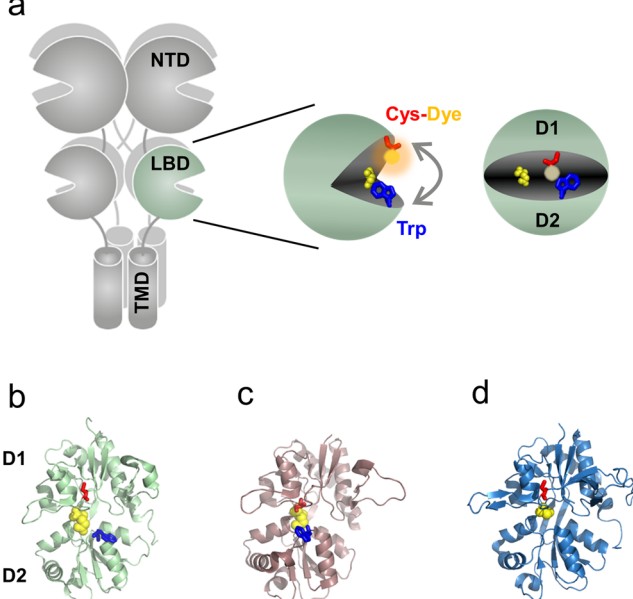

**Fig. 1 Fluorescence reporter design to detect clamshell motions of iGluR LBDs. a** Structural architecture of iGluRs and expanded view on the LBD, including the PET fluorescence reporter design for clamshell motions. In the bi-lobate LBD clamshell structure, the engineered Cys (red sticks) modified with fluorophore (orange sphere), and the engineered Trp (blue sticks) at the mouth of the clamshell, detect conformational dynamics (gray arrow) through contact-induced fluorescence quenching. The bound agonist is shown as yellow spheres and the upper D1 and lower D2 lobe is indicated. **b–d** Crystal structures of agonist-bound GluA2 (**b**), GluK1 (**c**), and GluN1 (**d**) LBDs in comic representation (PDB IDs: 2UXA, 1TXF, and 1PB7; for GluA2, GluK1, and GluN1, respectively). The structures are oriented as shown on the right-hand side of panel (**a**). Sites for fluorescence modification (engineered Cys) are shown as red sticks. Engineered Trp residues are shown as blue sticks. The bound agonist is highlighted as yellow spheres.

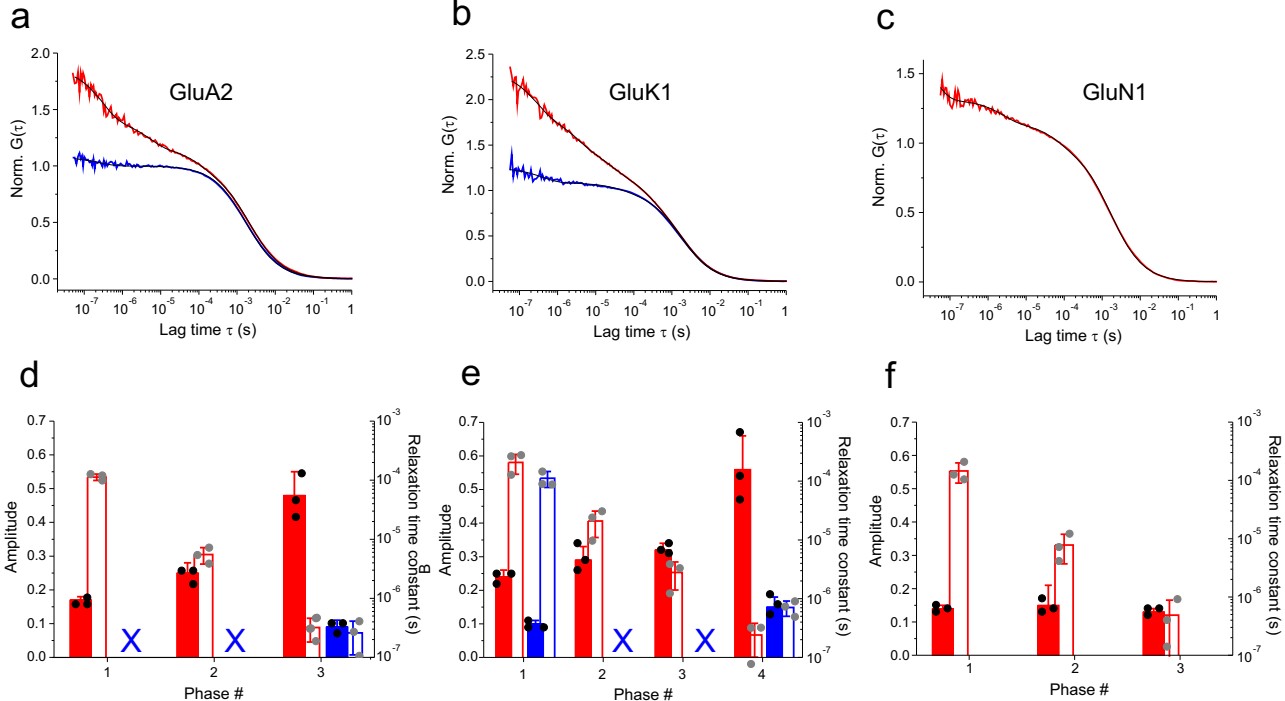

**Fig. 2 Clamshell dynamics of AMPA, kainate and NMDA iGluR LBDs in the apo state measured using FCS. a–c** ACFs ($G(\tau)$), normalized to the average number of molecules in the detection focus, recorded from GluA2 (**a**), GluK1 (**b**), and GluN1 (**c**) LBDs. ACFs of GluA2-LBD-G446C (**a**) and GluK1-LBD-K503C (**b**) are shown in blue. ACFs of GluA2-LBD-G446C-T685W (**a**) and GluK1-LBD-K503C-K734W (**b**) are shown in red. The ACF of GluN1-A480C is shown in red (**c**). Black lines are fits to the data using a model for molecular diffusion containing a sum of one to four single-exponential relaxations (indicated in (**d**–**f**)). **d**–**f** Amplitudes (closed bars) and corresponding time constants (open bars) obtained from fits to detected exponential phases in ACFs of the GluA2 (**d**), GluK1 (**e**) and GluN1 (**f**) LBD. Data sets and color code correspond to the ACFs shown in panels (**a**–**c**). Crosses (X) denote missing (not detected) kinetics. Error bars are s.d. of three measurements ($n = 3$).

sites in our reporter design resemble the structural coordinates chosen as measure of GluA2 LBD clamshell dynamics in previous molecular dynamics computer simulations[25]. Native Cys side chains in LBDs are either buried or form structural disulfides and thus did not interfere with site-specific modification of engineered Cys. This was evident from control experiments where modification trials of wild-type LBDs yielded only ~10% labeled protein whereas the LBDs containing the engineered Cys residues yielded ~60% labeled protein (Supplementary Table 1).

We performed FCS experiments using a confocal fluorescence microscope setup. For AMPA- and kainate-type GluA2 and GluK1 LBDs, the single-point (i.e. negative control) mutants G446C and K503C, modified with AttoOxa11, yielded a single decay of the autocorrelation function (ACF). The single decays in the ACFs were on the millisecond (ms) time scale and originated from fluctuations caused by Brownian diffusion of the LBDs through the detection focus. While ACFs from single-point mutants exhibited no additional fluorescence fluctuations on the sub-ms time scale, the GluA2 G446C-T685W and GluK1 K503C-K734W double-mutants showed pronounced sub-ms relaxations of substantial amplitude (Fig. 2a, b). These additional relaxations arose from D1 and D2 inter-lobe motions that lead to rapid formation and disruption of van der Waals contact between AttoOxa11 and the engineered W685 and W734 side chains. For the GluN1 NMDAR LBD, however, the fluorescently modified single-point mutant A480C showed sub-ms decays in the ACF even without engineered Trp residue in the lower D2 lobe (Fig. 2c). Testing various natural amino acids as potential quenchers in fluorescence experiments shows that oxazine fluorophores are substantially quenched by Trp, little quenched by tyrosine (Tyr, Y), but not quenched by phenylalanine (Phe, F)[32]. In the structure of the GluN1 LBD we

found W498/W731 and Y703/Y711 in the vicinity of labeling position A480C, residues which may quench fluorescence of AttoOxa11 (Supplementary Fig. 1a). We mutated these native Trp and Tyr residues individually to phenylalanine (Phe, F) to test their potential role in quenching fluorescence of AttoOxa11 at position A480C in FCS experiments. The mutation Trp to Phe is structurally conservative because it replaces an aromatic indole moiety by an aromatic benzene and eliminates potential quenching by Trp. We found that sub-ms relaxations were still present in the ACFs of the F mutants and hardly different compared to the ones detected for pseudo-wild-type mutant A480C. (Supplementary Fig. 1b, Supplementary Table 2). Results suggested that the fluctuations detected from GluN1-A480C resulted from changes in the micro-environment of the environmentally sensitive label, like polarity, that were mediated by LBD conformational motions.

The sub-ms relaxations evident in ACFs of the GluA2 and the GluN1 LBD were well described by a sum of three single-exponential decays (Fig. 2a, c, d, f). The relaxations of the GluK1 LBD required a sum of four single-exponential decays to describe them (Fig. 2b, e). A reduction of the number of exponentials in the applied fitting model lead to deviations of the fits from the data, while adding an additional exponential lead to either no significant improvement or over-fitting, which was evident from the appearance of physically unreasonable fitting parameters (Supplementary Fig. 2). Hence, we considered a three-exponential model in the description of data of GluA2 and GluN1 LBDs, and a four-exponential model in the description of data of GluK1 LBD most appropriate. Overall, we found that the modes of clamshell motion within this family of LBDs were on time scales of ~100 μs, ~10 μs and ~1 μs. We observed subtype specific variations of time constants and amplitudes (Fig. 2d–f).

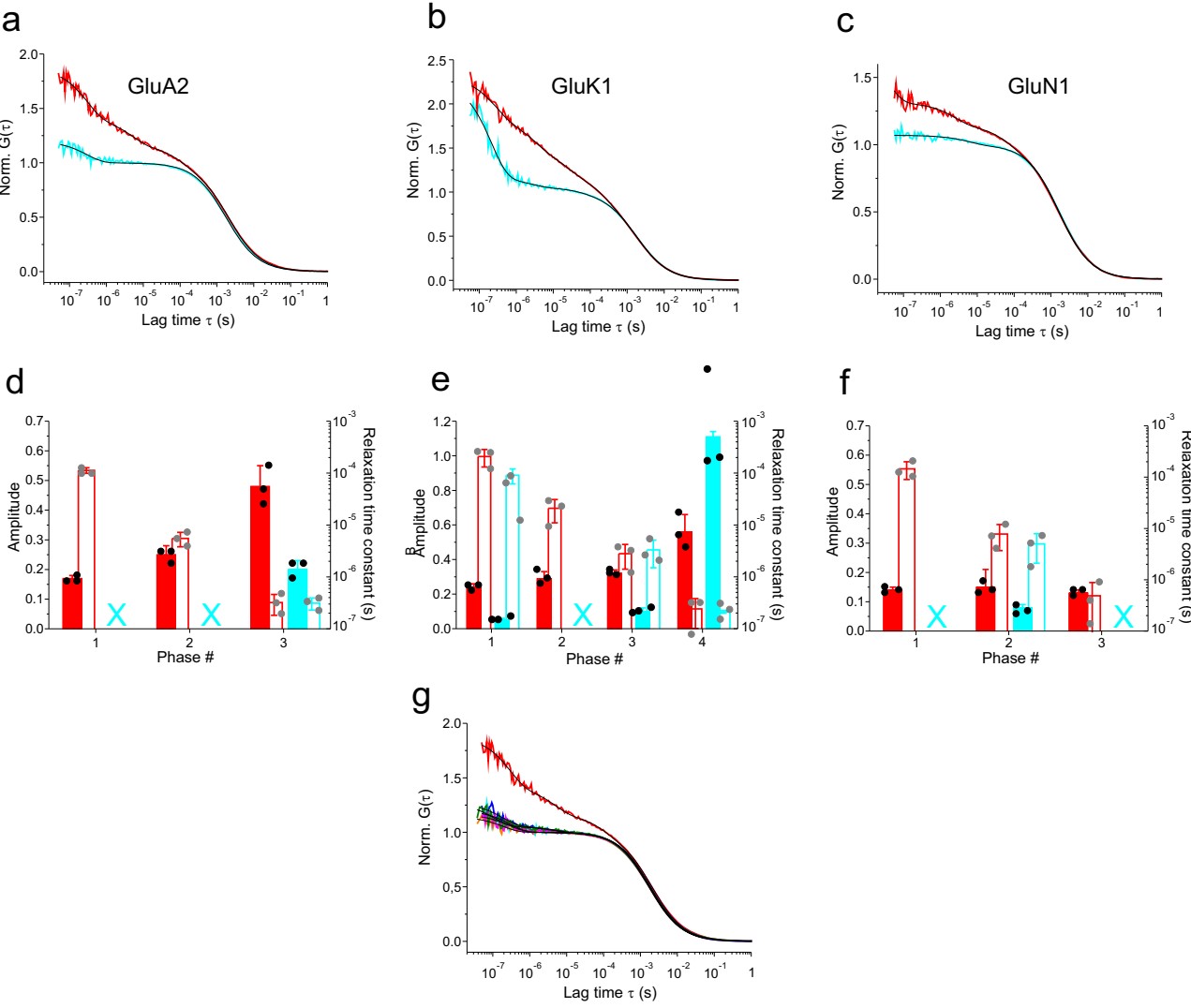

**Fig. 3 Influence of agonist binding on dynamics of iGluR LBDs. a–c** ACFs, (G(τ)), normalized to the average number of molecules in the detection focus, recorded from GluA2-LBD-G446C-T685W (**a**), GluK1-LBD-K503C-K734W (**b**), and GluN1-A480C (**c**) LBDs in the absence (red) and in the presence (cyan) of agonist. GluA2 and GluK1 LBDs had Glu as agonist. The GluN1 LBD had Gly as agonist. Black lines are fits to the data using a model for molecular diffusion containing a sum of one to four single-exponential relaxations (indicated in (**d–f**)). **d–f** Amplitudes (closed bars) and corresponding time constants (open bars) of exponential decays of ACFs of the GluA2 (**d**), GluK1 (**e**), and GluN1 (**f**) LBD. Data sets and color code correspond to the ACFs shown in panels (**a–c**). Crosses (X) denote missing (not detected) kinetics. Error bars are s.d. of three measurements (n = 3). **g** Normalized ACFs recorded from GluA2-LBD-G446C-T685W in absence (red) and in presence of quisqualate (blue), Glu (cyan), willardiine (magenta), 5-iodo-willardine (green), and kainate (orange). Black lines are fits to the data using a model for molecular diffusion containing a sum of one to three single-exponential relaxations.

**Binding of full and partial agonists stall clamshell motions.** Next, we investigated the influence of ligand binding on LBD clamshell dynamics. To this end, we applied saturating concentrations of agonist to fluorescently modified LBDs and recorded ACFs using FCS. L-glutamate (Glu) was applied as agonist for the AMPA- and kainate LBDs (constructs GluA2-G446C-T685W and GluK1-K503C-K734W), while glycine (Gly) was applied to the NMDA GluN1-A480C LBD. Glu and Gly did not directly influence fluorescence of the label AttoOxa11, which was evident from control FCS experiments where we applied the agonists to fluorescently modified constructs that did not contain the engineered Trp, or to the dye alone (Supplementary Fig. 3). For Trp-containing mutants we observed disappearance of sub-ms decays of the ACFs upon binding of agonist to the LBDs, which contrasted with the apo state that showed μs relaxations of substantial amplitude (Fig. 3). The kainate GluK1 LBD exhibited a loss of μs relaxations upon binding of Glu, similar as the other

homologues. But, by contrast to the homologues, the amplitude of the 200-ns kinetic phase increased (Fig. 3b, e). This indicated substantial mobility in the agonist-bound state of a presumably local structural element of the GluK1 LBD that fluctuated on the sub-μs time scale. Agonist-bound GluA2 and GluN1 LBDs exhibited residual fluctuations of minor amplitudes with corresponding time constants on the 300-ns and 5-μs time scale (Fig. 3d, f).

To exclude probe-induced artifacts and test for functionality of modified LBDs we performed ligand titration experiments where we added increasing concentrations of agonist to the fluorescently modified constructs. We used the magnitude of sub-ms fluctuation amplitudes as a measure for the bound/unbound state. The gradual losses of sub-ms amplitudes with increasing concentration of agonists were well described by a model for a binding isotherm (Supplementary Fig. 4). The analyses yielded equilibrium dissociation constants ($K_d$) for agonist binding that were within the range of values reported

in the literature. I.e., we found $K_d$(GluA2-LBD-G446C-T685W/Glu) = 2.4 ± 0.7 μM (literature[33–37]: $K_d$ = 0.2–1.7 μM), $K_d$(GluN1-A480C/Gly) = 13 ± 3 μM (literature[38]: $K_d$ = 26 μM), and $K_d$(GluK1-LBD-K503C-K734W/Glu) = 7 ± 3 μM. Minor discrepancies between our values and the literature values may be explained by the different solution conditions applied in the different experimental settings, or by fluorescence modification. We concluded that the investigated domains remained fully functional after modification.

Taken together our results showed that, within these iGluR LBD homologues, μs conformational motions of highly dynamic apo states were lost upon binding of agonists. This was apparent through either a complete loss of kinetic phases or by attenuated exponential signals of reduced decay amplitudes.

We next compared the influence of binding of full versus partial agonists[1] on clamshell dynamics of the GluA2 LBD. We applied saturating concentrations of the agonists quisqualate, L-Glu, willardiine, 5-iodo-willardiine, and kainate (listed in rank order of decreasing efficacy) to GluA2-G446C-T685W samples and recorded ACFs using FCS. We observed similar stalling of μs clamshell motions for all five agonists (Fig. 3g, Table 1). Kinetics of residual μs fluctuations of the GluA2 LBD in the presence of quisqualate and 5-iodo-willardiine had amplitudes of <10%, which we consider too small to reliably assign a motion. We thus did not detect notable differences in the modes of action of partial agonists compared to full agonists with regard to modulating conformational dynamics of the LBD.

**Dimerization slows clamshell motions of LBDs.** Within full-length iGluR assemblies, LBDs form dimers[9]. We investigated the influence of LBD dimerization on their clamshell dynamics. Isolated LBDs from AMPA and NMDA iGluR subtypes do not form dimers in solution up to protein concentrations of mg/ml[39,40]. Mutation L483Y in the dimer interface of the GluA2 LBD stabilizes a homo-dimer in solution ($K_d$ = 0.03 μM)[39]. Likewise, mutations N521Y and E516Y in the GluN1 and GluN2A LBD drive formation of specific hetero-dimers, i.e., formation of a GluN1-GluN2A LBD assembly[40]. We introduced mutation L483Y, N521Y, and E516Y in the GluA2, GluN1 and GluN2A LBD, respectively. Dynamics of homo-dimers formed by the GluA2 LBD were investigated by recording ACFs of GluA2-LBD-G446C-T685W-L483Y in presence of excess GluA2-LBD-L483Y (Fig. 4a). Dynamics of hetero-dimers formed by GluN1/GluN2A LBDs were investigated by measuring ACFs of GluN1-LBD-A480C-N521Y in presence of excess GluN2A-LBD-E516Y (Fig. 4b). The time constant of diffusion, $\tau_D$, of a globule scales directly with its hydrodynamic radius, $R_h$[41]. We determined values of $R_h$ for the Glu-bound GluA2-LBD monomer and dimer, as well as for the Gly-bound GluN1-LBD monomer and for the Gly/Glu-bound GluN1/GluN2A LBD hetero-dimer. We compared these values with the ones calculated from crystal structures (Table 2). $R_h$ of the GluA2 LBD, measured in solution using FCS, was significantly larger than the value calculated from the structure. This finding is in agreement with an ensemble of dynamic and more expanded LBD conformations in solution than in a crystal. This finding was also predicted by molecular dynamics simulations[25]. For LBD dimers we observed a ~30% increase of $R_h$ compared to the monomers (Table 2). This was as expected, because the doubling of molecular weight of a globule results in an $\sqrt[3]{2}$-fold (26%) increase of $R_h$[41].

We found that the pattern of sub-ms kinetics of LBD clamshell motions detected in apo monomers was preserved in dimers (Fig. 4c, d). However, the conformational motions slowed, which was evident from an increase of time constants of all three sub-ms relaxations upon dimerization. The corresponding amplitudes were reduced upon dimerization (Fig. 4c, d). Binding of agonist

stalled the μs conformational dynamics in dimers, similar as we observed for monomers, which was evident from the disappearance of μs relaxations (Fig. 4). We also applied the more efficacious agonist AMPA[26] to the GluA2 LBD. We found that changes of dynamics were essentially indistinguishable from the ones observed upon binding of Glu: we observed stalling of the two μs conformational relaxations. The residual nanosecond fluctuations that had similar amplitudes and time constants (Fig. 4c).

**Allosteric transmission of clamshell motions in NMDA LBDs.** NMDA LBDs form obligate hetero-dimers, contrasting with AMPA- and kainate-subtypes that form both homo- and hetero-dimers. In NMDARs, the GluN1 LBD binds glycine (Gly) while the GluN2A LBD binds L-glutamate (Glu), and isolated GluN1 and GluN2A LBDs can assemble into hetero-dimers in solution[40] (Fig. 4). This provided an opportunity to study allosteric communication between two LBDs, i.e. the effect that Glu-binding to the GluN2A LBD has on dynamics of the GluN1 LBD (Fig. 5a). We found that Gly, a full agonist of GluN1, stalled GluN1 LBD clamshell motions within the GluN1/GluN2A dimer, as expected. Interestingly, however, binding of Glu to GluN2A in the hetero-dimer stalled clamshell motions of the apo GluN1 LBD within the GluN1/GluN2A dimer (Fig. 5b, c). The observation revealed inter-dimer allostery. The allosteric effect was evident from the loss of the ~100-μs and ~10-μs kinetic phases of the apo GluN1 LBD upon binding of Glu to the GluN2A LBD within the dimer, similar as observed for the binding of Gly to the GluN1 LBD (Fig. 5c). In a control experiment we applied Glu to monomeric GluN1-LBD-A480C-N521Y and found no effect on clamshell motions (Fig. 5d). The result confirmed that the halt of motions of GluN1 induced by binding of Glu to the GluN2A LBD was indeed an allosteric effect and not induced by a direct interaction of Glu with the GluN1 LBD.

A specific tyrosine residue in the GluN1 LBD, Y535, located at the dimerization interface, plays an important role in modulating deactivation of the receptor[40]. To investigate the role of this residue in allostery of dynamics between the GluN2A and the GluN1 LBD, we generated the GluN1 mutant, Y535S, that removes the aromatic side chain[40] (construct GluN1-LBD-A480C-N521Y-Y535S). Mutation Y535S did not influence clamshell dynamics of the monomer, as was evident from modest changes of GluN1 LBD dynamics and preservation of stalling of motions upon binding of Gly (Supplementary Fig. 5). Kinetics of clamshell motions of GluN1/GluN2A LBD dimers, containing mutation Y535S (GluN1-Y535S/GluN2A), were also preserved (compare Fig. 5e, f and Fig. 4b, d), showing that this mutation did not perturb dynamics of the dimer. Moreover, application of Gly to the GluN1-Y535S/GluN2A dimer stalled the ~100-μs and ~10-μs kinetic phases of the GluN1 clamshell motion (Fig. 5e, f). This was the same effect as observed for the GluN1/GluN2A dimer without mutation Y535S (Fig. 5b, c). Interestingly, however, application of Glu to the GluN1-Y535S/GluN2A had virtually no effect on clamshell motions of GluN1 (Fig. 5e, f), which was in contrast to the behavior of GluN1/GluN2A without mutation Y535S (Fig. 5b, c). Therefore, deletion of the Y535 aromatic side chain uncoupled transmission of clamshell motions from the GluN2A to the GluN1 LBD, highlighting the critical role of Y535 in NMDAR LBD allosteric coupling.

**Discussion**
The wealth of structural studies on iGluRs support a model where the binding of agonist to the LBD stabilizes a closed-cleft conformation, a process that ultimately drives opening of the channel gate[24]. Structural studies show that the amplitude of lobe motion is on the order of one nanometer[28]. The fast millisecond kinetics[1]

**Table 1 Kinetics of sub-ms clamshell motions of GluA2-LBD-G446C-T685W in presence of partial and full agonists.**

|  | Apo | Quisqualate | Glu | Willardiine | 5-Iodo-willardine | Kainate |
|---|---|---|---|---|---|---|
| $a_1$ | 0.17 ± 0.01 | — | — | — | — | — |
| $\tau_1$ (µs) | 113 ± 14 | — | — | — | — | — |
| $a_2$ | 0.25 ± 0.03 | 0.06 ± 0.01 | — | — | 0.05 ± 0.01 | — |
| $\tau_2$ (µs) | 6 ± 2 | 4 ± 1 | — | — | 7 ± 1 | — |
| $a_3$ | 0.48 ± 0.07 | 0.22 ± 0.02 | 0.20 ± 0.03 | 0.17 ± 0.01 | 0.22 ± 0.01 | 0.15 ± 0.03 |
| $\tau_3$ (ns) | 320 ± 140 | 180 ± 50 | 300 ± 50 | 290 ± 100 | 116 ± 35 | 190 ± 40 |

Amplitudes ($a_n$) and corresponding time constants ($\tau_n$) are the mean of three measurements ($n = 3 \pm$ s.d.). (—) no kinetics observed.

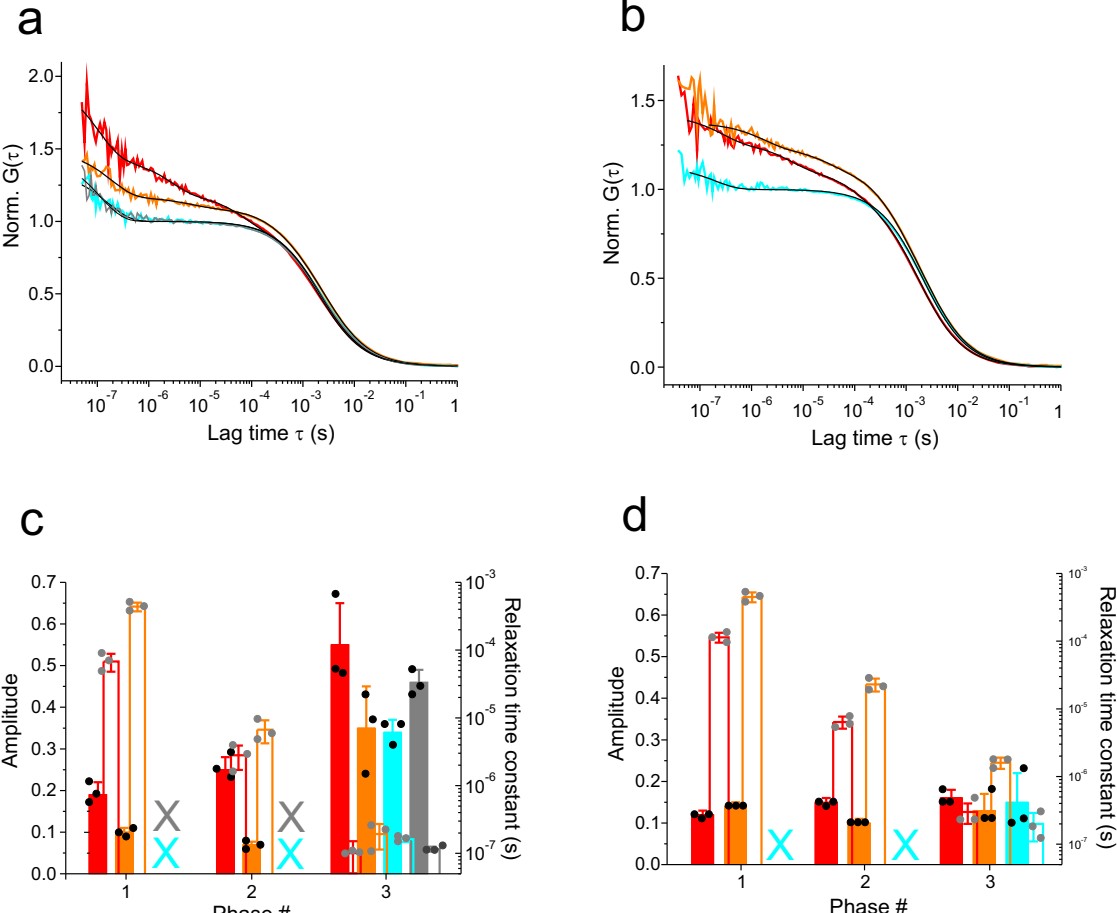

**Fig. 4 Clamshell dynamics of dimeric iGluR LBDs. a** ACFs, G($\tau$), normalized to the average number of molecules in the detection focus, recorded from GluA2-LBD-G446C-T685W-L483Y (apo monomer, red) and in presence of excess GluA2-LBD-L483Y (apo dimer, orange). ACFs of the dimer in presence of the agonists Glu and AMPA are shown in cyan and gray (agonist-bound dimers). **b** ACFs recorded from GluN1-LBD-A480C-N521Y (apo monomer, red) and in presence of excess GluN2A-LBD-E516Y (apo hetero-dimer, orange). The ACF of the GluN1-GluN2A hetero-dimer in presence of the agonist Gly is shown in cyan. Black lines are fits to the data using a model for molecular diffusion containing a sum of one to three single-exponential decays (indicated in (**c**, **d**)). **c** Amplitudes (closed bars) and corresponding time constants (open bars) of exponential decays of ACFs of the monomeric and dimeric GluA2 LBD in apo and agonist bound states. Data sets and color code correspond to the ACFs shown in panel (**a**). Crosses (X) denote missing (not detected) kinetics. **d** Amplitudes (closed bars) and corresponding time constants (open bars) of exponential decays of ACFs of the monomeric and hetero-dimeric GluN1 LBD in apo and agonist-bound states. Data sets and color code correspond to the ACFs shown in panel (**b**). Crosses (X) denote missing (not detected) kinetics. Error bars are s.d. of three measurements ($n = 3$).

of receptor activation suggest that the underlying LBD clamshell motions should be on a similar time scale. Clamshell motions of the isolated domains can be expected to be even faster because inter- and intra-subunit interactions of domains within full-length iGluR assemblies will slow dynamics. Here, we employed nanosecond FCS in combination with an environmentally sensitive

oxazine label to probe LBD dynamics on the one-nanometer spatial and sub-ms temporal scale. We found pronounced multi-exponential kinetics of LBD reconfiguration in the apo state, which had substantial relaxation amplitudes. Since a two-state conformational transition follows a mono-exponential time course[42], the number of sub-ms exponentials detected by FCS can

**Table 2 Hydrodynamic radii of AMPA and NMDA LBDs measured using FCS and compared with values calculated from available x-ray structures.**

|  | GluA2, monomer, apo | GluA2, monomer, Glu | GluA2, dimer, apo | GluA2, dimer, Glu |
|---|---|---|---|---|
| $R_h$(x-ray) (nm) | 2.5 | 2.4 | 3.1 | 3.0 |
| $R_h$(FCS) (nm) | 2.9 ± 0.1 | 2.7 ± 0.1 | 3.8 ± 0.2 | 3.3 ± 0.2 |
|  | GluN1, monomer, apo | GluN1, monomer, Gly | GluN1/N2A, dimer, apo | GluN1/GluN2A, dimer, Gly/Glu |
| $R_h$(x-ray) (nm) | 2.6 | 2.5 | — | 3.2 |
| $R_h$(FCS) (nm) | 2.6 ± 0.2 | 2.4 ± 0.1 | 3.5 ± 0.3 | 3.2 ± 0.4 |

PDB IDs of structures: 1FTJ (GluA2 LBD monomer/dimer); 1PB7 (GluN1 LBD monomer); 2A5T (GluN1/GluN2A LBD dimer)).
Values of $R_h$ are mean values of three measurements ($n = 3 \pm$ s.d.).—structural data not available.

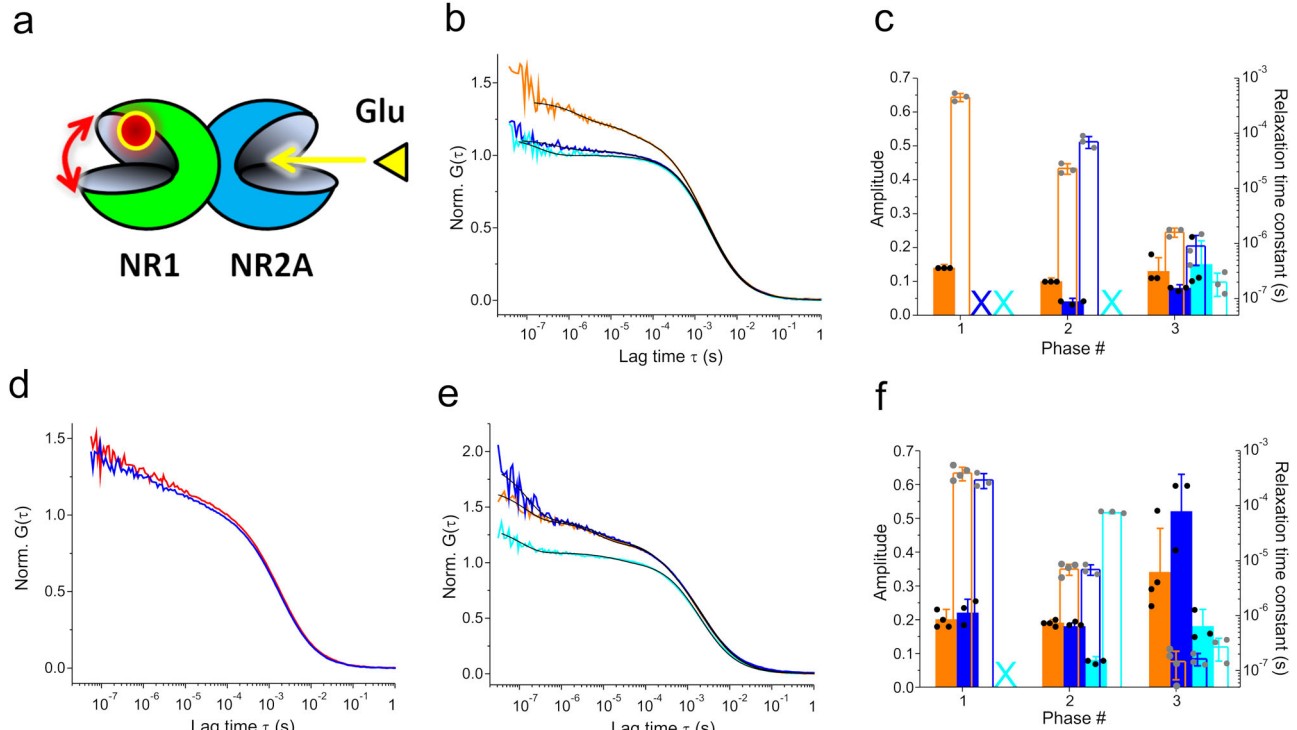

**Fig. 5 Allosteric communication of clamshell dynamics in NMDA iGluR LBD dimers. a** Design of the reporter system that senses effects of agonist binding to the GluN2A LBD on dynamics of the GluN1 LBD. The fluorescence label on the GluN1 LBD is illustrated by a red sphere and LBD clamshell dynamics by a red arrow. Binding of agonist Glu is indicated by a yellow arrow. **b** ACFs, G(τ), normalized to the average number of molecules in the detection focus, recorded from the GluN1 LBD within the GluN1/GluN2A dimer (orange data, GluN1-LBD-A480C-N521Y/GluN2A-LBD-E516Y). ACFs of the hetero-dimer measured in presence of the agonist Glu or Gly are shown in blue and cyan, respectively. Black lines are fits to the data using a model for molecular diffusion containing a sum of one to three single-exponential decays (indicated in panel (**c**)). **c** Amplitudes (closed bars) and corresponding time constants (open bars) of exponential decays of ACFs recorded from the GluN1/GluN2A dimer. Data sets and color code correspond to ACFs shown in panel (**b**). Crosses (X) denote missing (not detected) kinetics. **d** ACFs of the monomeric apo GluN1-LBD (construct A480C-N521Y) measured in absence (red) and in presence (blue) of 1 mM Glu. **e** ACF of the GluN1/GluN2A LBD dimer containing mutation Y535S (construct GluN1-LBD-A480C-N521Y-Y535S/GluN2A-LBD-E516Y), measured in the apo state (orange), in the presence of Glu (blue), or in the presence of Gly (cyan). Black lines are fits to the data using a model for molecular diffusion containing a sum of two or three single-exponential decays (indicated in panel (**f**)). **f** Amplitudes (closed bars) and corresponding time constants (open bars) of exponential decays of ACFs recorded from the GluN1/GluN2A LBD dimer containing mutation Y535S. Data sets and color code correspond to the ACFs shown in panel (**e**). The cross (X) denotes missing (not detected) kinetics. Error bars are s.d. of three measurements ($n = 3$).

be interpreted as the number of modes of conformational change. For AMPA and NMDA subtype LBDs we detected three modes of motions that were on the ns-to-μs time scale, whereas the kainate LBD showed four modes of motion. FCS measures equilibrium kinetics and a single-exponential decay of the ACF can be interpreted as originating from one thermally activated conformational change along a specific reaction coordinate. In protein dynamics,

the rank order of time scales of conformational changes generally follows the rank order of spatial scales on which they occur. Loop motions and movement of secondary structure are on the ns-to-μs time scale while collective domain motions occur on the μs-to-ms time scale[43]. Molecular dynamics simulations carried out on NMDA iGluR LBDs predict a rich spectrum of lobe motions[44]: three principal components were identified in the simulations and

classified as hinge-bending, sweeping and twisting. Intriguingly, the number of principal components identified in simulations matches the number of exponentials detected in our experiments, suggesting that the same processes are observed. Simulations further show expanded conformations sampled by the LBD beyond of what is observed in structural studies[25,44]. smFRET spectroscopy detects a heterogeneous ensemble of conformations of the GluA2 LBD that interconvert on the ~100-ms time scale[45]. This time scale is more than three orders of magnitude slower than the ones we detected here using FCS. The discrepancy suggests that the nature conformational transitions detected by smFRET and FCS are different. smFRET in combination with fast correlation spectroscopy, however, detects rapid inter-domain dynamics between the two upper D1 lobes within a dimeric LBD assembly from a metabotropic glutamate receptor (mGluR)[46]. The 50-100 μs time scale detected for inter-domain D1-D1 motions agrees with time constants detected here for intra-subunit D1-D2 lobe motions of iGluR LBDs. Interestingly, the rapid D1-D1 dynamics were detected in the ligand-bound active and resting states of mGluR LBD assemblies, showing that the ligand-bound domains remain mobile with respect to each other[46].

We found that the binding of agonists stalled sub-ms clamshell motions of all three iGluR LBD homologues, which was evident from the disappearance of exponential decays in the ACFs (Fig. 3). Rapid, thermally activated equilibrium fluctuations in the apo states detected by FCS suggest a conformational-selection mechanism of ligand-binding. This has also been inferred from results of previous sub-ms smFRET spectroscopy carried out on NMDA LBDs[47]. We observed residual conformational fluctuations in agonist-bound states of LBDs (Figs. 3, 4). These occurred mainly on the ultrafast sub-μs time scale. Molecular dynamics simulations predict residual flexibility in the agonist-bound states of AMPA and NMDA LBDs[23,25,44], in agreement with our results. NMR studies find evidence for flexibility both in the apo and agonist-bound states[48].

We detected sub-μs fluctuations of pronounced amplitude in the agonist bound state of the GluK1 LBD. The amplitude was substantially higher compared to amplitudes of the corresponding decays seen in GluA2 and GluN1 LBDs. The finding may be explained by the fact the fluorophore labeling site is part of an extended loop segment of GluK1, which is two residues longer compared to the same segments in GluA2 and GluN1 and may thus provide additional flexibility (Fig. 1c).

We found that the binding of the full agonists AMPA and Glu had virtually indistinguishable effects on LBD dynamics: both agonists stalled the same modes of motion of the GluA2 LBD, both in the monomeric and dimeric states. (Fig. 4c). The observation can be explained by the similarity of crystal structures of the GluA2 LBD in complex with Glu or AMPA[24], which suggests similar modes of action. Binding of partial agonists to iGluRs reduces the amplitudes of electrophysiological signals compared to the binding of full agonists, but the mechanism behind is not yet fully understood[49]. Binding of partial agonists may lead to a more open LBD clamshell conformation compared to the binding of full agonists and may consequently trigger the channel gate with lower probability. This scenario is suggested by structural studies on a series of sterically demanding, substituted willardiines bound to the GluA2 LBD that show gradually more open domains[49]. smFRET experiments carried out on the GluA2 LBD show a heterogeneous ensemble of structural states populated upon binding of willardiines[50], which seem to dial different conformations. An alternative model of partial agonism suggests that the lower stability of the LBD/partial-agonist complex reduces efficacy despite inducing a fully closed clamshell[51]. A third scenario may involve higher mobility of the LBD bound to a partial agonist compared to bound to a full agonist, which could

lead to lower probability of triggering the channel gate. Here we found that binding of partial agonists, like substitute willardiines, to the GluA2 LBD yielded identical dynamic signatures compared to the binding of full agonists, like quisqualate, AMPA or Glu, (Fig. 3g, Table 1). The result supports a mechanism whereby different stabilities or selected conformations of LBDs in complex with full or partial agonists are responsible for different efficacies.

We observed that dimerization of LBDs resulted in an increase of the time constants of clamshell motions and a decrease of the corresponding kinetic amplitudes (Fig. 4). This is reasonable because the dimerization interface covers the hinge region of the clamshell. Dimerization will thus restrict mobility of the D1 and D2 lobes with respect to each other.

Assembly of GluN1 and GluN2A LBDs as hetero-dimers allowed us to study allosteric effects. We investigated the effect that the binding of Glu to the GluN2A LBD had on clamshell dynamics of the GluN1 LBD within the dimer. To our surprise, we found that binding of Glu to the GluN2A LBD stalled motions of GluN1, similar as Gly did by directly binding to GluN1 (Fig. 5). It was hypothesized previously that the binding of an agonist to one LBD subunit may stabilize a closed-lobe state of a neighboring unliganded subunit[52,53]. This is exactly what we observed here. We further found that mutation Y535S, located in the hinge region of the GluN1 LBD, abolished the allosteric effect: binding of Glu to the GluN2A LBD within the GluN1/GluN2A LBD dimer, where the Y535 aromatic side chain was deleted, had virtually no effect on dynamics of the GluN1 LBD (Fig. 5). The side chain of Y535 in the GluN1/GluN2A dimerization interface fills a pocket that is the target of allosteric modulators of the homologous AMPA iGluR LBD[40]. Electrophysiology shows that mutation Y535S accelerates deactivation of NMDA iGluRs and is proposed to serve as a clutch between GluN1 and GluN2A LBDs[40]. This proposal is in agreement with our findings, showing that Y535 transmits the ligand-binding induced stalling of dynamics from the GluN2A LBD to the GluN1 LBD. smFRET studies carried out on full-length receptors propose enhanced conformational spread and flexibility of the GluN1 LBD induced by binding of Glu to GluN2A. Conclusions are based on spread and broadening of smFRET histograms recorded from the GluN1 LBD within a receptor upon application of Glu[54]. The discrepancy between smFRET and our FCS results may be explained by the different spatial and temporal scales probed by the methods. FRET probes global conformational changes on the 2–10-nm scale while contact-induced quenching is active on the 1-nm scale[55]. The conformational states observed in smFRET histograms interconvert slower than ms, which is inherent to the applied method of data acquisition, while FCS detects fast dynamics on the time scale of ns-ms[31]. It is thus likely that different conformational sub-states are probed by the different techniques. Moreover, since smFRET experiments were carried out within the context of the full-length receptor[54], inter- and intra-subunit interactions of labels within the receptor may modulate their fluorescence emission intensities and contribute to heterogeneity of smFRET histograms. Molecular dynamics simulations carried out in the same study, however, show that binding of Glu to the GluN2A LBD induces a more closed conformation of the GluN1 LBD cleft[54], which is in agreement with stalled GluN1 LBD motions observed here.

Negative cooperativity of Gly with Glu in NMDA iGluRs is reported[2,54,56]. The binding of Glu to GluN2A lowers the affinity of GluN1 to Gly[40]. Our results suggest a mobility mechanism behind the phenomenon: binding of Glu to the GluN2A LBD stalls dynamics of the GluN1 LBD via an allosteric pathway involving Y535. Within the framework of agonist binding through conformational selection, reduced flexibility of the GluN1 LBD attenuates its affinity to Gly.

## Methods

**Protein mutagenesis, synthesis, and fluorescence modification**. DNA constructs contained the genes encoding *Rattus norvegicus* AMPA GluA2, kainate GluK1, NMDA GluN1, and NMDA GluN2A ligand binding domain lobes S1 and S2, connected by Gly-Thr linker, and a N-terminal His$_6$-tag, as part of T7 expression vectors pET22b(+) for the GluA2, GluK1, and GluN1 construct, and pET22b(+)-Sumo for the GluN2A construct[40]. Single-point mutants were generated using the QuikChange mutagenesis protocol (Stratagene).

LBDs and mutants thereof were overexpressed in Origami 2 (DE) (AMPA constructs) or Origami B (DE3) (kainate and NMDA constructs) *Escherichia coli* cells (Novagen) using the T7 expression system. After growing bacterial cells to an OD$_{600nm}$ of 2.0 in liquid Terrific Broth (TB) medium (AMPA constructs) or Luria-Bertani (LB) medium (kainate and NMDA constructs), containing 100 µg/ml ampicillin (AMPA constructs) or 50 µg/ml ampicillin and 12.5 µg/ml tetracycline (kainate and NMDA constructs), overexpression was induced by adding 0.4 mM (AMPA constructs) or 0.5 mM (kainate and NMDA constructs) isopropyl β-D-1-thiogalactopyranoside (IPTG, Sigma-Aldrich). Origami B cells were then incubated at 15 °C for 20 h, whereas Origami 2 cells were incubated at 18 °C for 20 h. After lysis using sonication His$_6$-tagged protein was isolated from bacterial cell lysates by loading them onto a Talon®Superflow™ chromatography column (Sigma-Aldrich) equilibrated in 20 mM Tris-HCl, pH 8.0, 300 mM NaCl, 10 mM imidazole, containing 1 mM L-glutamic acid potassium salt monohydrate (Sigma-Aldrich), or glycine (Sigma-Aldrich) (kainate and NMDA constructs), or 20 mM Tris-HCl, pH 8.0, 500 mM NaCl, 20 mM imidazole, containing L-glutamic acid potassium salt monohydrate (AMPA constructs). In the case of kainate and NMDA constructs, Talon®Superflow™ resin was washed by applying a gradient of 16 mM to 32 mM imidazole in equilibration buffer. LBDs were eluted from the resin using 20 mM Tris-HCl, pH 8.0, 300 mM NaCl, containing 250 mM imidazole and 1 mM L-glutamic acid potassium salt monohydrate or glycine (kainate and NMDA constructs), or 20 mM Tris-HCl, pH 8.0, 500 mM NaCl, containing 500 mM imidazole and 1 mM L-glutamic acid potassium salt monohydrate (AMPA constructs). The eluate was loaded onto a size exclusion chromatography (SEC) column (HiLoad 26/60 Superdex™ 75) (GE Healthcare) in 20 mM HEPES, 150 mM NaCl, pH 7.5, containing 1 mM L-glutamic acid potassium salt monohydrate (AMPA constructs), or, for kainate and NMDA constructs, dialyzed into a buffer for anion exchange chromatography (20 mM Tris-HCl, pH 8.5, 20 mM NaCl, containing 1 mM L-glutamic acid potassium salt monohydrate or glycine). During preparation of NMDA GluN2A LBD constructs, the sumo fusion protein was removed through proteolytic digestion (sumo protease Ulp-1). In case of AMPA GluA2, kainate GluK1, and NMDA GluN1 LBD, the His$_6$-tag was removed by proteolytic digestion using thrombin (Sigma-Aldrich). LBDs were further purified using anion exchange chromatography (5 ml HiTRap™ Q FF column, GE Healthcare) applying a gradient from 20 mM to 500 mM NaCl in 20 mM Tris-HCl, pH 8.5, containing 1 mM L-glutamic acid potassium salt monohydrate or glycine. Pooled fractions containing LBD were concentrated using a 10-kDa-MWCO centrifugal concentrator (Sartorius™ Vivaspin™ 20). Purity of LBDs was confirmed using SDS-PAGE.

Single-point Cys mutants of LBD constructs were modified using the thiol-reactive maleimide derivative of the dye AttoOxa11 (Atto-Tec). Labeling was carried out in 20 mM HEPES, 150 mM NaCl, pH 7.5, containing 1 mM L-glutamic acid potassium salt or glycine, and a tenfold molar excess of the reducing agent tris(2-carboxyethyl)phosphine (TCEP, Sigma-Aldrich). A 15-fold molar excess of AttoOxa11 over LBD was applied. The labeling reaction was carried out for 3 h at 4 °C. Excess dye was removed using SEC on a Sephadex G-25 column (GE Healthcare).

The degree of labeling (DOL) of LBDs was determined from the UV-absorption signal of the protein and the Vis-absorptions signal of the label of the conjugate, following the manufacturer's instructions (Atto-Tec):

$$\mathrm{DOL} = \frac{c(\mathrm{dye})}{c(\mathrm{protein})} = \frac{A_{\max} \cdot \varepsilon_{\mathrm{prot}}}{(A_{280} - A_{\max} \cdot CF_{280}) \cdot \varepsilon_{\max}} \tag{1}$$

where $A_{\max}$ is the absorption signal of the conjugate at the wavelength maximum of the dye, $\varepsilon_{\mathrm{prot}}$ is the extinction coefficient of the protein at 280 nm, $A_{280}$ is the absorption signal of the conjugate at 280 nm, $CF_{280}$ is the dye-specific correction factor of the label that corrects for the contribution of the dye, and $\varepsilon_{\max}$ is the extinction coefficient of the label at the wavelength maximum.

**FCS experiments**. Nanosecond FCS was carried out on a custom-built confocal fluorescence microscope setup, applying cross-correlation of signals from two fiber-coupled avalanche photodiode detectors (Perkin Elmer, SPCM-AQRH-15-FC) to bypass detector dead-time and after-pulsing effects, using a digital hardware correlator device (ALV 5000/60×0 multiple tau digital real correlator)[57]. LBD constructs were measured in 50 mM phosphate, pH 7.5, with the ionic strength adjusted to 200 mM using potassium chloride. GluN1/GluN2A LBD constructs were measured in 20 mM HEPES, pH 7.5, with the ionic strength adjusted to 150 mM using sodium chloride. 0.05 % Tween-20 (Sigma-Aldrich) and 0.3 mg/ml bovine serum albumin (BSA, Sigma-Aldrich) were used as buffer additives to suppress glass surface interactions. ACFs were recorded from approx. 1 nM fluorescently modified LBD samples. For experiments involving LBD dimerization, an excess of typically 40 µM of unlabeled LBD construct was added to the fluorescently modified LBD construct. In agonist-binding studies, 1 mM agonist was added to samples prior to measurement. All measurements were performed at 25 °C. Accurate control of temperature was achieved using a custom-built objective heater. All samples were filtered using a 0.2-µm syringe filter and transferred onto a high-precision coverslip before measurements. For each experimental setting, three ACFs from one sample were recorded ($n = 3$) of 10 minutes duration each.

**Data analysis**. ACFs were analyzed using analytical models[58] and the parameters of interest were extracted from the fits to the data. The analytical model for two-dimensional diffusion of a globule through the detection volume was applied[57]:

$$G(\tau) = \frac{1}{N}\left(1 + \frac{\tau}{\tau_D}\right)^{-1} \tag{2}$$

N is the average number of molecules in the detection focus, and $\tau_D$ represents the time constant of diffusion. The application of a model for diffusion in two dimensions was of sufficient accuracy because the two horizontal dimensions $(x, y)$ of the detection focus were much smaller than the lateral dimension $(z)$ in the applied setup.

ACFs containing additional decays on the sub-ms time scale were described by a sum of single-exponentials using an extended analytical model[57]:

$$G(\tau) = \frac{1}{N}\left(1 + \frac{\tau}{\tau_D}\right)^{-1}\left(1 + \sum_n a_n \exp\left(-\frac{\tau}{\tau_n}\right)\right) \tag{3}$$

where $a_n$ denotes the amplitude and $\tau_n$ the time constant of the $n$th decay.

In this model, a change between a fluorescent and a fluorescence-quenched conformation is described by a two-state equilibrium, the kinetics of which follow a mono-exponential decay. The amplitude and time constant of the exponential contain the microscopic rate constants of the on/off fluorescence fluctuation. Several independent two-state conformational relaxations are described by a sum of exponentials. A microscopic analysis of rate constants is complicated by that fact that the amplitude of the decay is also modulated by the brightness of states[31].

Values of $\tau_D$ were converted into $R_h$ using the Stokes–Einstein equation[41]. For calibration of the setup we measured $\tau_D$ of the fluorophore Atto655 as a reference, for which the diffusion coefficient and thus $R_h$ is known[59]. For comparison, radii of gyration ($R_g$) of LBDs were calculated from available crystal structures using PyMOL (pymol.org). Values of $R_h$ were calculated from $R_g$ using the theoretical scaling law for a monodisperse sphere[60]:

$$R_h = \frac{1}{\sqrt{3/5}} \cdot R_g \tag{4}$$

For determination of $K_d$ values from the FCS relaxation amplitudes plotted versus concentration of agonist, we applied the model for a protein-ligand binding isotherm (P + L = PL)[61]:

$$[PL] = \frac{[P]_t[L]}{[L] + K_d} \tag{5}$$

where $[PL]$ is the concentration of the protein-ligand complex, $[P]_t$ is the total concentration of fluorescently modified protein, and $[L]$ is the concentration of the ligand. The change of the FCS relaxation amplitude $a$ was modeled as:

$$\frac{a - a_u}{a_b - a_u} = \frac{[L]}{[L] + K_d} \tag{6}$$

where $a$ is the observed amplitude, $a_u$ is the amplitude in the unbound state, and $a_b$ is the amplitude in the bound state.

**Reporting summary**. Further information on research design is available in the Nature Research Reporting Summary linked to this article.

## Data availability
The data that support the findings of this study are available from the corresponding author on reasonable request. Source data of figures are available as Supplementary Data.

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

## Acknowledgements

The authors thank Hiro Furukawa for the provision of plasmids for heterologous expression of GluN1 and GluN2A LBDs. This publication was supported by the Open Access Publication Fund of the University of Würzburg.

## Author contributions

S.R., S.S., I.H.G. and H.N. designed experiments. S.R, S.S., L.B. and A.P. performed experiments. S.R, S.S and L.B. analyzed data. S.R., A.P., I.H.G., and H.N. wrote the paper.

## Funding

## Competing interests

The authors declare no competing interests.
