## [Transparent Peer Review File · Communications Biology]

Reviewers' comments:

Reviewer #1 (Remarks to the Author):

In this paper, Rajab et al describes conformational dynamics of isolated clamshell-shaped ligand-binding domains from the three major iGluR sub-types, using PETFCS. I like the paper, but point 1 is really crucial and authors must have to address the point.

1) The author need to show, experimentally, that binding of dyes does not have any effect on structure and function of the proteins. Without this experiment, the biological importance of paper is incomplete.

2) The authors should show the details of fitting. That means the residual distribution of the fittings should be shown. Additionally they need to show, how much the residual distribution is changing if they increase or decrease the number of components.

3) The author should remove that "design photoinduced electron transfer probes" unless they made the fluorescent probe. They use the statement again by saying that "designed fluorescence probes that..." but it look like that they did not design the probe.

4) A general description of 'amplitude' and 'time constants' will be useful for general readers. The authors can include the supporting information.

Reviewer #2 (Remarks to the Author):

In the presented paper by Rajab et al. the authors use PET-FCS to study the dynamics of closure of the LBDs of the three major ionotropic glutamate receptor subtypes. They analyze the dynamics of single LBDs as well as LBD homo and heterodimers with nanosecond resolution and in response to partial and full agonists. Finally, they study the mechanism of allostery in the NMDA heterodimeric LBDs through monitoring the changes in dynamics in one LBD in response to Glu binding to the other LBD. Overall, the authors find μ s dynamics that slow down and/or disappear upon ligand binding.

Although the dynamics of the iGluR LBDs have been studied using different methods before I find that this study provides a very good complement. In particular the addressed timescales are of high relevance for conformational rearrangements of such domains and especially in the context of the timescales of activation of iGluRs by Glu at synapses. Furthermore PET allows addressing motions of shorter range than FRET thus providing additional information to what has been done before.

Overall, the manuscript is well written and the argumentation of the authors can easily be followed including sufficient information to make it accessible to non-specialists. I therefore recommend this article for publication after a couple of minor points have been addressed.

1) The authors state that they did not observe labeling of native cysteines although some are present in the LBDs. I would much appreciate if this was supported by data in the supplementary.

2)The authors note that the GluN1 NMDAR LBD shows subms decays even without engineered Trp in D2 lobe. They further identify potential residues including Trp and Tyr that may be responsible for quenching of their probe and mutate these to Phe which does not abolish the observed fluctuations. What is the reasoning for mutating to Phe and not to any other amino acid residue?

3) Figure 1b-d are not mentioned in the text.

4) The PET measurements presented rely on the quenching/changes in the microenvironment of a

single fluorophore and thus the capability of drawing conclusions on conformational changes and dynamics strongly depends on appropriate controls. While the authors do a very good job in providing such controls, I feel that an analysis of the influence of ligands on the probe in LBDs lacking the Trps described to lead to quenching would further reinforce the conclusions.

5) In the end of the discussion the authors mention the discrepancy in allosteric modulation of the NMDA heterodimer LBDs between their study supported by MDS and previous smFRET measurements. The authors should elaborate a bit more on that point and e.g. suggest potential drawbacks/uncertainties of each of the methods, try to provide a potential biological/structural (free LBDs over full-length receptors) explanation or at least make a proposition how this discrepancy could be addressed in the future.

Response to the referee comments on COMMSBIO-21-0974

We thank both referees for their time and effort reviewing our manuscript and for their valuable comments. In the following, we address the comments (cited in *italics*) point by point and highlight changes made in the revised manuscript.

Reviewer 1:

Comment:

In this paper, Rajab et al describes conformational dynamics of isolated clamshell-shaped ligand-binding domains from the three major iGluR sub-types, using PETFCS. I like the paper, but point 1 is really crucial and authors must have to address the point.

1) The author need to show, experimentally, that binding of dyes does not have any effect on structure and function of the proteins. Without this experiment, the biological importance of paper is incomplete.

Response:

We thank the reviewer for the positive assessment of our work.

We agree with the reviewer that probe-induced artefacts must be excluded in order to draw reasonable biological conclusions. There are several experimental observations in our study that show that the investigated LBDs retain their function after fluorescence modification. (i) Conformational fluctuations of LBDs, probed by the label, stall upon binding of agonist, which is evident from vanishing sub-ms decays in ACFs and substantially reduced fluctuation amplitudes. This shows that the modified LBDs respond to agonist and are thus functional. (ii) In dimerization experiments of LBDs, hydrodynamic radii (R_h) obtained from FCS data of fluorescently modified LBDs indeed increase to the values expected for a globule of twice the molecular weight (described and discussed on page 12-13 of the revised manuscript; values of R_h shown in Table 2), as expected. This shows that fluorescently modified LBDs are capable of forming dimers, which are observed in structural studies of isolated LBDs and full-length receptors. (iii) Specificity of binding is confirmed by a control experiment where addition of L-glutamate (Glu) to the fluorescently modified GluN1 LBD (which responds to glycine and not to glutamate) did not at all influence its dynamics. As expected, addition of the agonist glycine (Gly) did stall the conformational fluctuations (Figures 3c and 5c).

We nevertheless agree with the reviewer that additional quantitative measures are required to test for and confirm a lack of probe-induced artefacts. We therefore performed additional ligand titration experiments. We monitored binding of agonists to fluorescently modified LBDs by the loss of sub-ms fluctuation amplitudes of conformational relaxations. In titration experiments, the observed loss of relaxation amplitude at increasing concentration of agonist fitted well to a binding isotherm. Fits to the data yielded equilibrium dissociations constants (K_d) that were within the range of values reported in the literature. I.e., we found $K_d(\text{GluA2-LBD-G446C-T685W/Glu}) = 2.4 \pm 0.7 \mu\text{M}$ (literature: refs. 31-35: $K_d = 0.2-1.7 \mu\text{M}$), $K_d(\text{GluN1-A480C/Gly}) = 13 \pm 3 \mu\text{M}$ (literature: ref. 36: $K_d = 26 \mu\text{M}$) and $K_d(\text{GluK1-LBD-K503C-K437W/Glu}) = 7 \pm 3 \mu\text{M}$. Discrepancies between values of K_d may be explained by the different solution conditions applied in the different experimental settings or protein modification. We concluded that the investigated domains remained fully functional after modification. The new ligand titration experiments and corresponding results are described on page 9 of the revised manuscript. Binding isotherms are provided as new Supplementary Figure 4. Analysis of binding isotherms

is described in the Methods section.

Comment:

2) The authors should show the details of fitting. That means the residual distribution of the fittings should be shown. Additionally they need to show, how much the residual distribution is changing if they increase or decrease the number of components.

Response:

We fully agree with the reviewer that the complex kinetics observed requires measures that justify the applied models. We now show residual plots of fits to the ACFs recorded from the GluA2-, GluK1 and GluN1-LBD constructs GluA2-LBD-G446C-T685W, GluK1-LBD-K503C-K734W, and GluN1-A480C, which contain varying numbers of exponentials. A reduction of the number of exponentials in the fitting model leads to deviations of the fits from the data, while adding an additional exponential lead to either no significant improvement or over-fitting, which was evident from physically unreasonable fitting parameters. The fits to the data together with the corresponding residual plots are shown in a new Supplementary Figure 2. We refer to this analysis on page 7 of the revised manuscript.

Comment:

3) The author should remove that “design photoinduced electron transfer probes” unless they made the fluorescent probe. They use the statement again by saying that “designed fluorescence probes that...” but it look like that they did not design the probe.

Response:

The reviewer is correct. In contrast to labels on GluA2 and GluK1 LBD, the label on the GluN1 LBD does not represent a PET fluorescence probe because it does not involve quenching by Trp. We therefore removed “design photoinduced electron transfer probes” from the abstract and replaced the phrase by “design fluorescence probes”. The word “design” refers to the fact that the position of label is not chosen arbitrarily but tailored to report a change of clamshell conformation by a change of fluorescence emission intensity.

Comment:

4) A general description of ‘amplitude’ and ‘time constants’ will be useful for general readers. The authors can include the supporting information.

Response:

In the applied FCS model, a change between a fluorescent and a fluorescence-quenched conformation is described by a two-state equilibrium, the kinetics of which follow a mono-exponential decay. The amplitude and time constant of the exponential contain the microscopic rate constants of the on/off fluorescence fluctuation (described in ref. 28 of the revised manuscript). The presence of several independent two-state conformational changes is described by a sum of exponentials. However, a microscopic analysis is complicated by the fact

that the amplitude of the decay is also modulated by the brightness of states (ref. 28). We added the description of the model to the Methods section on page 27 of the revised manuscript.

Reviewer 2:

Comment:

In the presented paper by Rajab et al. the authors use PET-FCS to study the dynamics of closure of the LBDs of the three major ionotropic glutamate receptor subtypes. They analyze the dynamics of single LBDs as well as LBD homo and heterodimers with nanosecond resolution and in response to partial and full agonists. Finally, they study the mechanism of allostery in the NMDA heterodimeric LBDs through monitoring the changes in dynamics in one LBD in response to Glu binding to the other LBD. Overall, the authors find μ s dynamics that slow down and/or disappear upon ligand binding.

Although the dynamics of the iGluR LBDs have been studied using different methods before I find that this study provides a very good complement. In particular the addressed timescales are of high relevance for conformational rearrangements of such domains and especially in the context of the timescales of activation of iGluRs by Glu at synapses. Furthermore PET allows addressing motions of shorter range than FRET thus providing additional information to what has been done before.

Overall, the manuscript is well written and the argumentation of the authors can easily be followed including sufficient information to make it accessible to non-specialists. I therefore recommend this article for publication after a couple of minor points have been addressed.

Response:

We thank the reviewer for the positive assessment of our work.

Comment:

1) The authors state that they did not observe labeling of native cysteines although some are present in the LBDs. I would much appreciate if this was supported by data in the supplementary.

Response:

We agree with the reviewer and provide the degrees of labelling (DOLs) of our constructs in new Supplementary Table 1 of our revised manuscript. The determination of the DOL from the absorption signals of label and protein are described in the Methods section of the revised manuscript. Wild-type LBDs yielded ~10% labelled protein whereas LBDs containing engineered Cys residues yielded ~60% labelled protein. We refer to the DOLs on page 5 of the revised manuscript.

Comment:

2) The authors note that the GluN1 NMDAR LBD shows subms decays even without engineered Trp in D2 lobe. They further identify potential residues including Trp and Tyr that may be

responsible for quenching of their probe and mutate these to Phe which does not abolish the observed fluctuations. What is the reasoning for mutating to Phe and not to any other amino acid residue?

Response:

The AttoOxa11 label applied in our study is the same as the better known commercial Atto655 chromophore, but contains a shorter alkyl linker between the thiol-reactive group and the chromophore. Testing various natural amino acids as quenchers in fluorescence experiments shows that the fluorophore is substantially quenched by Trp, little quenched by Tyr, but not at all quenched by Phe (ref. 30 in the revised manuscript). The mutation Trp to Phe belongs to the class of non-disruptive deletion mutations. It is structurally conservative because it replaces an aromatic indole moiety by an aromatic benzene. We applied this mutation to test for potential quenching effects of native Trp, aiming at minimal structural or energetic perturbation induced by mutation. We added this explanation to page 6 of the revised manuscript.

Comment:

3) Figure 1b-d are not mentioned in the text.

Response:

We thank the reviewer for spotting this mistake. We now refer to Figure 1b-d on pages 4 and 5 of the manuscript.

Comment:

4) The PET measurements presented rely on the quenching/changes in the microenvironment of a single fluorophore and thus the capability of drawing conclusions on conformational changes and dynamics strongly depends on appropriate controls. While the authors do a very good job in providing such controls, I feel that an analysis of the influence of ligands on the probe in LBDs lacking the Trps described to lead to quenching would further reinforce the conclusions.

Response:

We agree with the reviewer that the influence of the ligands on the label should be investigated in additional control experiments. We therefore performed additional measurements where we recorded PET-FCS data from constructs GluA2-LBD-G446C and GluK1-LBD-K503C, which do not contain the engineered Trp, in presence and absence of L-glutamate. Since GluN1-A480C, which does not include an engineered Trp, was effective in reporting conformational dynamics by fluorescence fluctuations, we recorded FCS data from the dye alone in absence and presence of glycine, to probe the influence of this agonist on the label. We did not observe any significant changes of the ACFs upon addition of 1 mM ligand, which confirms that the ligands do not influence fluorescence of the label. Some residual fluctuations of the GluK1-K503C-LBD that vanished upon application of 1 mM Glu are explained by the pronounced dynamics of this LBD (compare Figure 2b) that appear to be picked up by the label alone, but

to a very minor extent. We added these data as new Supplementary Figure 3 to the revised manuscript and discussed them on page 9 of the revised manuscript.

Comment:

5) In the end of the discussion the authors mention the discrepancy in allosteric modulation of the NMDA heterodimer LBDs between their study supported by MDS and previous smFRET measurements. The authors should elaborate a bit more on that point and e.g. suggest potential drawbacks/uncertainties of each of the methods, try to provide a potential biological/structural (free LBDs over full-length receptors) explanation or at least make a proposition how this discrepancy could be addressed in the future.

Response:

We agree with the reviewer that the discrepancy between results obtained from smFRET spectroscopy (ref. 52 in the revised manuscript) and PET-FCS calls for a more extended discussion. We added the following paragraph to the Discussion section on page 21-22 of the revised manuscript: “The discrepancy between the FRET and PET fluorescence results may be explained by the different spatial and temporal scales probed by both methods. FRET probes global conformational changes on the 2-10-nm scale while PET is active on the 1-nm scale (ref. 53 in the revised manuscript). The conformational states detected in smFRET histograms interconvert slower than ms, which is inherent to the applied method of data acquisition, while PET-FCS detects fast dynamics on the time scale of ns-ms (ref. 28 in the revised manuscript). It is thus likely that different conformational sub-states are probed by the different techniques. Moreover, since smFRET experiments were carried out within the context of the full-length receptor, inter- and intra-subunit interactions of labels within the receptor may modulate their fluorescence emission intensities and contribute to heterogeneity of smFRET histograms.”

REVIEWERS' COMMENTS:

Reviewer #1 (Remarks to the Author):

The authors have successfully addressed my comments.

Reviewer #2 (Remarks to the Author):

The authors have sufficiently addressed all my comments in the revised manuscript. I recommend the revised version for publication.